# Dose-Dependent Effects of a Novel Selective EP_4_ Prostaglandin Receptor Agonist on Treatment of Critical Size Femoral Bone Defects in a Rat Model

**DOI:** 10.3390/biomedicines9111712

**Published:** 2021-11-18

**Authors:** Corina Vater, Elisabeth Mehnert, Henriette Bretschneider, Julia Bolte, Lisa Findeisen, Lucas-Maximilian Matuszewski, Stefan Zwingenberger

**Affiliations:** 1University Center of Orthopedic, Trauma and Plastic Surgery, University Hospital Carl Gustav Carus, Fetscherstrasse 74, 01307 Dresden, Germany; Henriette.Bretschneider@uniklinikum-dresden.de (H.B.); Julia.Bolte@uniklinikum-dresden.de (J.B.); Lisa.Findeisen@tu-dresden.de (L.F.); Lucas.Matuszewski@uniklinikum-dresden.de (L.-M.M.); Stefan.Zwingenberger@uniklinikum-dresden.de (S.Z.); 2Center for Translational Bone, Joint and Soft Tissue Research, Medical Faculty and University Hospital Carl Gustav Carus, Technische Universität Dresden, Fetscherstrasse 74, 01307 Dresden, Germany

**Keywords:** critical-sized bone defect, bone regeneration, tissue regeneration, scaffold, EP_4_ receptor agonist, BMP-2

## Abstract

Difficulties in treating pseudarthrosis and critical bone defects are still evident in physicians’ clinical routines. Bone morphogenetic protein 2 (BMP-2) has shown promising osteoinductive results but also considerable side effects, not unexpected given that it is a morphogen. Thus, the bone regenerative potential of the novel selective, non-morphogenic EP_4_ prostaglandin receptor agonist KMN-159 was investigated in this study. Therefore, mineralized collagen type-1 matrices were loaded with different amounts of BMP-2 or KMN-159 and implanted into a 5 mm critical-sized femoral defect in rats. After 12 weeks of observation, micro-computed tomography scans were performed to analyze the newly formed bone volume (BV) and bone mineral density (BMD). Histological analysis was performed to evaluate the degree of defect healing and the number of vessels, osteoclasts, and osteoblasts. Data were evaluated using Kruskal-Wallis followed by Dunn’s post hoc test. As expected, animals treated with BMP-2, the positive control for this model, showed a high amount of newly formed BV as well as bone healing. For KMN-159, a dose-dependent effect on bone regeneration could be observed up to a dose optimum, demonstrating that this non-morphogenic mechanism of action can stimulate bone formation in this model system.

## 1. Introduction

The difficulty in treating disturbed or delayed fracture healing is still evident in the daily routine of physicians. Trauma, pseudarthrosis, infection, and tumor excision can lead to critical bone defects that require bone replacement. Autologous bone transplantation is the gold standard in treating critical bone defects today [1]. Since autologous bone transplantation is not feasible to an unlimited extent, and because it involves a number of risks and disadvantages, new possibilities for bone replacement and osteopromotive or osteoinductive materials are being sought in various directions. Therefore, growth factors for improved bone regeneration are being investigated and used clinically [2]. The most important are the bone morphogenetic proteins (BMPs). With the exception of BMP-1, a metalloproteinase, they represent a subgroup of the transforming growth factor-β (TGF-β) family and contribute to the differentiation of mesenchymal stromal cells (MSCs) that are important for bone healing and other development-relevant processes in the body [3,4]. In the case of bone formation, they support the differentiation of osteoblasts and osteogenesis just as they activate osteoclasts for bone resorption. However, it has been shown that strong ectopic bone formation can occur due to BMPs’ morphogenic activity, resulting in, for example, nerve damage to the spinal column. Sexual dysfunction, as well as bowel and bladder dysfunction, have also been reported [5,6,7].

Prostaglandin E_2_ (PGE_2_), a tissue hormone derived from arachidonic acid, has also been shown to have a promoting effect on bone formation and healing via its PGE_2_ receptors 2 and 4 (E-type prostanoid (EP) receptors 2/4) [8]. It is involved in angiogenesis and the differentiation of various cells of bone remodeling [9]. Furthermore, prostaglandins act as inflammatory mediators and thus play an important role as initiators in the inflammatory phase at the onset of bone healing [10,11].

Synthetically produced PGE_2_ has been used in other disciplines for a long time [12,13,14,15]. However, due to severe systemic side effects (lethargy, diarrhea, flush syndrome), prostaglandins are not approved for the treatment of skeletal diseases and critical bone defects. Since it has been known that PGE_2_ mainly acts on the skeleton via the G protein-coupled receptors EP_2_ and EP_4_, such receptor agonists have been the subject of many studies. For example, another EP_4_ agonist, ONO-AE1-329, has already been shown to improve bone healing in animal models [16,17]. Cayman Chemical (Ann Arbor, MI, USA) developed a series of small molecule difluorolactam EP_4_ receptor agonists, including KMN-159, the EP_4_ receptor agonist used in this study. It exhibits high selectivity and efficacy at the EP_4_ receptor [18], is chemically stable, rapidly cleared from the organism through urine, and has simpler storage conditions than BMP-2 [19].

We hypothesized that the EP_4_ receptor agonist KMN-159 would demonstrate a significant bone regenerative potential in a critical-sized femoral bone defect model in rats without the possibility of the side effects that can occur with the use of a morphogen such as BMP-2.

## 2. Materials and Methods

### 2.1. Study Design

In this study mineralized collagen matrix (MCM) scaffolds were loaded with different concentrations of rhBMP-2 or the EP_4_ prostaglandin receptor agonist KMN-159. Subsequently, the functionalized scaffolds were implanted into 5 mm critical-sized femoral bone defects in rats. After an observation time of 12 weeks, the animals were euthanized, and femora were explanted and prepared for µCT scans and histological analyses.

### 2.2. Preparation of the Implants

Mineralized collagen matrix (MCM) scaffolds were produced as described previously [20] and sterilized by supercritical CO_2_ [21] (15 µg KMN group) or by gamma irradiation (all other groups; minimum dose: 25 kGy). Then, scaffolds were loaded with 200 µL (maximum loading volume of the MCM scaffolds determined in advance) containing either a high dose (50 µg) rhBMP-2 (InductOS^®^, Medtronic BioPharma, Heerlen, The Netherlands) in saline or with 15 µg, 200 µg, 2 mg, 5 mg, or 10 mg KMN-159 (Cayman Chemical, Ann Arbor, MI, USA) in ethanol under sterile conditions using a laminar airflow bench. To ensure proper binding of the BMP-2 and the KMN-159, scaffolds were allowed to dry overnight under the laminar airflow bench. Following drying, scaffolds were stored at 4 °C until use (KMN-159) or used the same day for implantation (BMP-2). The flexible, sponge-like cylindrical MCM scaffolds had a length of 8 mm and a diameter of 4 mm.

### 2.3. Animals

For the in vivo study, 64 male, 12–13 weeks old RjHan-WI Wistar rats (weight: 386.7 ± 27.2 g, range 340–456 g) were purchased from Janvier Labs (Le Genest-Saint-Isle, Mayenne department, France) and randomized into 6 groups (Table 1). While kept at 2 animals/cage at a 12-h light-and-dark-cycle, animals were fed a standard diet with food and water ad libitum. All animal experiments were performed in accordance with the National Institutes of Health Guidelines for the Use of Experimental Animals and were approved by the Local Animal Care and Ethics Committee of Dresden University Hospital (protocol no: DD24.1-5131/354/10, 29 April 2016).

### 2.4. Surgical Procedure

Animals were anesthetized using a mixture of 2.5–3% isoflurane and oxygen maintained at a flow rate of 1.5–2 L/min (isoflurane vaporiser: Ohmeda-Isotec 3, BOC Health Care, Great Britain, UK). For analgesia, buprenorphine was injected subcutaneously at a concentration of 30 µg/kg.

Surgeries were performed in prone position as described previously [22]. Briefly, the right hind limb of the animal was shaved and disinfected, and an approximately 3 cm long skin and fascia incision was made along the length of the right femur. After presenting and cleaning the right femur laterally from soft tissue, a customized 5-hole internal fixation plate (Ø 1.5 mm straight locking plate, stainless steel; LCP Compact Hand System, Synthes GmbH, Oberdorf, Switzerland) was fixed to the femur using 2 proximal and 2 distal screws (Ø 1.5 mm locking screws, stainless steel; length outer screws: 7 mm, length inner screws: 6 mm; Synthes GmbH). A custom-made 3-dimensional printed saw guide was then placed on the lateral femur above the third empty hole of the plate, and a 5 mm osteotomy was performed using 2 Gigli wires (0.44 mm; RISystem AG, Landquart, Switzerland). The piece of mid-diaphyseal bone was carefully removed, and the wound was thoroughly rinsed with saline to remove metallic remnants of the Gigli wires. The 5 mm defect was then filled press-fit with the group-specific MCM scaffold. Soft tissue was then returned to its anatomical position, and the wound was closed in a layered fashion (muscles, fascia, skin) using resorbable sutures (Vicryl 4-0, Ethicon, Johnson & Johnson, New Brunswick, NJ, USA). After wound closure, animals were transferred back into their cage and allowed to wake up under an infrared lamp.

### 2.5. Preparations for Micro-Computed Tomography (µCT) and Histology

After an observation period of 12 weeks, animals were anaesthetized with isoflurane and then sacrificed by exposure to CO_2_ following cervical dislocation. Right and left femora were explanted and cleaned from soft tissue. After that, the osteosynthesis screws and the plate were carefully removed from the right femora. Particular care was taken not to injure the developed callus. Samples were then transferred to 15 mL tubes filled with 4% neutral buffered formaldehyde that was changed every 48 h and stored at 4 °C until further analysis. Under these conditions, µCT scanning of all femora was performed. Formaldehyde was then replaced by ethylenediaminetetraacetic acid (EDTA) to decalcify the bones. The EDTA was changed every 48 h over a period of 14 days. After decalcification, the bones were dehydrated by using an ascending alcohol series and then embedded in paraffin. Samples were then sagittally cut into 2 µm thick slices. Subsequently, hematoxylin-eosin (HE), actin, tartrate-resistant acid phosphatase (TSP), and alkaline phosphatase (ALPL) stains were performed.

### 2.6. High-Resolution µCT Analysis

The µCT analysis was performed using a SCANCO vivaCT 40 (SCANCO Medical AG, Wangen-Brüttisellen, Switzerland) with the following settings: X-ray intensity = 114 µA, X-ray tube = 70 kVp, voxel size = 21 µm, integration time = 200 ms. The evaluation was conducted using the SCANCO vivaCT software. For each sample, a region of interest (ROI) was determined in the center between the 2 inner screw holes. The ROI consisted of 380 slices covering a length of 7.98 mm. A bone mineral density (BMD) of 200 mg hydroxyapatite/cm^3^ was set as the lower limit for analysis, and bone volume (BV) was measured in mm^3^.

### 2.7. Histological Examination

The grade of defect healing, according to Huo et al. [23], was classified by 3 independent, blinded observers based on HE stainings (Merck, Darmstadt, Germany). Therefore, 3 representative sections of each femur were analyzed.

Vascularization was analyzed based on α-smooth muscle actin staining (rabbit anti-smooth muscle actin, clone 1A1, 1:750, Cat.# M0851, Agilent Dako, Santa Clara, CA, USA). All vessels in the defect area showing a lumen were counted.

To evaluate the activity of osteoblasts, the slides were stained with bone alkaline phosphatase (BAP, anti-rabbit IgG peroxidase, 1:100, Cat.# PAK0142, LINARIS Biologische Produkte GmbH, Dossenheim, Germany), whereas the activity of osteoclasts was analyzed by tartrate-resistant acid phosphatase staining (TSP, Sigma-Aldrich, St. Louis, MO, USA). All stained cells within the defect area fulfilling the morphological requirements for osteoblasts and osteoclasts were counted. While osteoblasts are polygonal or cuboidal mononuclear cells [24], osteoclasts appear as multinuclear cells with a ruffled border and a diameter up to 100 µm [25]. The evaluation of all histological sections was performed using a Keyence BIOREVO BZ-9000 microscope (Keyence, Neu-Isenburg, Germany).

### 2.8. Statistics

All values are presented as mean value ± standard deviation and 95% confidence interval (mean ± SD, 95% CI). Statistical analysis was performed by using GraphPad Prism 8 (GraphPad Software, San Diego, CA, USA) with setting the level of significance to *p* = 0.05. As the data were not normally distributed, the Kruskal–Wallis test was used to examine statistical significance. Differences between the groups were evaluated using Dunn’s post hoc test.

## 3. Results

A 5 mm critical mid-diaphyseal defect was created in 12–13 weeks old RjHan-WI Wistar rats, stabilized with an osteosynthesis plate, and augmented by an MCM scaffold functionalized with rhBMP-2 or increasing amounts of KMN-159. Defect healing was evaluated at 12 weeks post-surgery. 62 out of 64 animals survived the surgeries and the observation period. One animal of the 2 mg, 5 mg, and 10 mg KMN-159 group, respectively, died intra-, 1 h or 2 days postoperatively.

Compared to animals treated with rhBMP-2 and low doses of KMN-159, rats treated with high doses of KMN-159 (5 mg and 10 mg) showed an unusually long recovery phase after surgery as well as lethargy, “fluffed” fur, and secretion from the Harderian gland for about 3 days post-surgery. These side effects could be managed by repetitive analgesia with buprenorphine until the symptoms disappeared. However, three animals also showed a massive overgrowing of the osteosynthesis plate with new bone, which was not observed for the other groups.

### 3.1. µCT Analysis

After an observation period of 12 weeks, µCT analysis was performed to investigate the volume of the newly regenerated bone and its mineral density (Table 2, Figure 1 and Figure 2).

Statistical analysis showed a significant influence of the drug loaded onto the scaffold on the formation of new bone (*p* = 0.0001). Thereby, a significantly higher bone volume (BV) for the 50 µg rhBMP-2 group compared with the 15 µg KMN-159 (*p* = 0.0001), 200 µg KMN-159 (*p* = 0.0028) and 5 mg KMN-159 group (*p* = 0.0037) was observed. Interestingly, no significant difference regarding the newly formed BV could be detected between rhBMP-2 and 2 mg (*p* = 0.1700) and 10 mg KMN-159 (*p* = 0.5829).

Accordingly, bone mineral density (BMD) was also significantly influenced by the type of scaffold (*p* = 0.0001). Highest BMD values could be observed for the low and middle dose KMN-159 groups which was significant when compared to 50 µg rhBMP-2 (15 µg: *p* < 0.0001; 200 µg: *p* < 0.01; 2 mg: *p* < 0.05). For the 2 high dose KMN-159 groups (5 mg and 10 mg), BMD values were comparable to that of 50 µg rhBMP-2.

### 3.2. Histological Analysis

Figure 3 shows representative examples of all groups of the HE- and the immunohistochemical actin staining. Representative examples of the TSP- and immunohistochemical ALPL-staining are shown in Figure 4.

#### 3.2.1. Histological Grade of Defect Healing

The grade of defect healing was determined based on the score established by Huo et al. [23] (Figure 5, Table 3). Consistent with the bone volume results, the drug loaded onto the scaffold significantly influenced the grade of defect healing (*p* < 0.0001). The lowest degree of defect healing was observed for animals treated with 15 µg KMN-159 (equivalent to previous negative controls) and the 200 µg KMN-159 (*p* < 0.01). The 2 mg KMN-159 and the 10 mg KMN-159 groups (both *p* < 0.0001) all showed significantly increased bone healing in comparison. As expected, treatment with a high dose (50 µg) rhBMP-2 led to the highest degree of defect healing (= defect bridging with mature bone), which was significant compared to all other groups (*p* < 0.0001).

#### 3.2.2. Histological Analysis of Vascularization and Bone Cell Markers

In line with BV and the grade of defect healing, there was a significant influence of the drug loaded onto the implant regarding vascularization and the number of osteoblasts and osteoclasts (all *p* < 0.001, Table 3).

The number of vessels in the defect area (Figure 6A) was highest for 15 µg KMN-159 which was significant compared to 50 µg rhBMP-2 (*p* < 0.0001), 200 µg KMN-159 (*p* = 0.0372) and 2 mg KMN-159 (*p* = 0.0045). Animals treated with 50 µg rhBMP-2 showed the lowest degree of vascularization, followed by middle and high dose KMN-159. Compared to the number of vessels found in 15 µg KMN-159 samples, they were reduced to 12–15% when middle dose KMN-159 (200 µg, 2 mg) and to 19–22% when high dose KMN-159 (5 mg, 10 mg) was applied. The lowest number of vessels was detected in rhBMP-2 treated samples.

The number of osteoblasts (Figure 6B) was inversely correlated to the concentration of KMN-159. So, the highest cell numbers were detected in the defect area of animals treated with 15 µg KMN-159 followed by 200 µg, 2 mg, 5 mg, and 10 mg KMN-159. The lowest numbers of osteoblasts were found in the 50 µg rhBMP-2 group. Therefore, significant differences were detected between 15 µg KMN-159 and 50 µg rhBMP-2 (*p* < 0.0001), 5 mg (*p* = 0.0001) and 10 mg KMN-159 (*p* < 0.0001) as well as between 50 µg rhBMP-2 and 200 µg KMN-159 (*p* < 0.05). While in the 15 µg KMN-159 group osteoblasts were mainly found within the defect area, for all other groups, osteoblasts were mainly seen at the bony ends.

As for osteoblasts, the highest numbers of osteoclasts (Figure 6C) were found in the 15 µg KMN-159 group followed by 5 mg and 10 mg KMN-159, 50 µg rhBMP-2, 2 mg and 200 µg KMN-159. Therefore, significant differences were found between 200 µg KMN-159 and 15 µg KMN-159 (*p* = 0.0005), 5 mg KMN-159 (*p* = 0.0486) and 10 mg KMN-159 (*p* = 0.0328).

## 4. Discussion

Currently, rhBMP-2 is the only Food and Drug Administration (FDA) approved growth factor for bone regeneration in the market. However, apart from its proven osteoregenerative effect, its use can also be associated with severe side effects due to its morphogenic activity. The aim of the current study was to evaluate the suitability of the novel selective EP4 prostaglandin receptor agonist KMN-159 to act as a non-morphogenic alternative to rhBMP-2. Thus, mineralized collagen scaffolds were loaded with five different concentrations of KMN-159 or rhBMP-2, implanted into 5 mm critical-sized femoral defects in rats and analyzed after 12 weeks of observation regarding bone regeneration using radiographic and histological methods.

As expected, the use of rhBMP-2 led to complete bridging of the defect in all operated animals with the highest volume of newly formed bone and the highest degree of defect bridging. However, bone mineral density was below the contralateral control and lowest among all other groups, which is expected, based on other preclinical and clinical studies [26]. Additionally, in contrast to its high osteoinductive success in rodents, BMP-2 fails in up to 33% of human patients because of patient-specific gene expression profiles modulating the response to BMP-2 [27]. These results, coupled with its high cost and stringent storage conditions, limit its widespread use in dental and orthopedic applications. Thus, one potential therapeutic alternative to BMP-2 could be small molecules that activate alternative bone anabolic pathways.

Prostaglandin E_2_ (PGE_2_) governs a wide variety of physiological effects in many tissues and is mediated by signaling through four distinct E-type prostanoid (EP) receptors, of which EP_2_ and EP_4_ are important for bone [28]. The osteoinductive potential of EP_4_ agonists has been proven in various studies [17,29]. For KMN-159, a selective difluorolactam EP_4_ receptor agonist developed by Cayman Chemical, stimulation of osteogenesis as verified by cell culture studies and increased bone healing in a rat calvarial defect model has been shown [18,19]. In the present study, we observed a dose–dependent increase in bone volume and the degree of defect healing up to 2 mg of KMN-159, but no further increase by using higher KMN-159 concentrations. Thereby, complete defect bridging could be seen in none of the animals of the 15 µg and 200 µg KNM-159 group, but in 3/9 animals of the 2 mg and in 1/10 animals of the 5 mg and 10 mg KMN-159 group, respectively. Interestingly, regarding the volume of newly formed bone, there was no significant difference between rhBMP-2 and 2 mg and 10 mg KMN-159, respectively. In line with this, Owen et al. showed comparable healing rates of rat calvarial defects after five weeks when treated with either 330 ng KMN-159 or 550 ng BMP-2 [19]. Despite comparable bone volume and degree of defect healing, the use of high dose KMN-159 (5 mg and 10 mg) in our study led to adverse side effects like lethargy and secretion from the Harderian gland in the first days after surgery. This has also been reported for other EP_4_ agonists [17] and might be explained by the fact that the EP_4_ receptor is also involved in regulating pain and inflammation and that EP_4_ receptor antagonists are currently in development as pharmaceuticals to control pain and inflammation associated with osteoarthritis in veterinary patients [30].

Since bone regeneration and remodeling is already completed in rhBMP-2-treated animals after 12 weeks, only low numbers of vessels, osteoblast, and osteoclasts were seen within the defect area, as verified by histological analyses. In contrast, strong vascularization and activity of osteoblasts and osteoclasts could be detected for 15 µg KMN-159, indicating an ongoing remodeling process. This is supported by large residues of the mineralized collagen scaffold material that are still present in the defect area of 15 µg and 200 µg KMN-159-treated animals. In summary of the findings, we could show a dose–dependent osteoregenerative effect of KMN-159 with reaching its maximum at 2 mg KMN-159. Higher concentrations did not lead to increased defect healing, probably due to the dual role of prostaglandins in bone dynamics shifting the balance to a more resorptive state and inducing adverse side effects.

One possibility to enhance bone regeneration for low and middle dose KMN-159 is to optimize the carrier material. The mineralized collagen scaffold used in our study fully releases KMN-159 in a burst fashion within approximately 30 min, whereas calcium phosphate cements show a sustained release of KMN-159 for up to 7 days [31]. As shown by Owen et al., even a short exposure to KMN-159—as for the burst release from the mineralized collagen scaffold in our study—is sufficient to support osteogenesis [19]. Thus, only 10 min of exposure led to a 3-fold increase in the activity of alkaline phosphatase (ALP) when rat bone marrow cells were incubated with KMN-159 in vitro. Albeit more prolonged incubation did not yield any additional increase in ALP activity, the expression of Runt-related transcription factor 2 (RUNX2) was significantly increased at days 19 and 25, potentially enhancing the expression of late bone markers like osteocalcin. Thus, a sustained release of KMN-159 from the implant carrier material might be beneficial to stimulate and enhance bone defect regeneration. Additionally, the combination with other bioactive factors might also enhance bone regeneration by acting synergistically and thereby reducing the concentrations needed.

As with most studies, the design of the current study is subject to some limitations. First, bone regeneration was investigated after 12 weeks only. Since KMN-159 was released from the scaffold within minutes after implantation, it would be beneficial to analyze bone formation at an earlier time point. This would allow deeper insights into the dynamics of bone regeneration and a better assessment of the potential of KMN-159. Investigation of implant or scaffold degradation at a later time point might be of interest as well. The second limitation concerns the µCT measurement for which the steel-made osteosynthesis plate and screws had to be removed to avoid artefacts. In contrast to all other groups, osteosynthesis plates from animals treated with 5 and 10 mg KMN-159 exhibited massive overgrowth by new bone. Thus, to get the plate and screws out of the femur, the newly formed bone had to be dissected and was therefore missed in the analysis of bone volume and bone mineral density, affecting the overall statistical conclusions regarding bone formation in these two dose groups. This observation of robust bone growth is significant and suggests that if the mild side effects observed with these higher doses of KMN-159 can be overcome through a modified delivery mechanism, a complete bridging of the defects may occur. To overcome this shortfall, osteosynthesis plates made of polyetheretherketone (PEEK) are an alternative due to their radiographic transparency. Another limitation lies in the relatively high concentration of 50 µg rhBMP-2 that was used as positive control and for comparison with KMN-159. Being the carrier material, the minimum effective dose for bridging a 6–8 mm segmental femoral bone defect in rats is between 1 and 2.25 µg [32,33]. So, by using low-dose rhBMP-2 as a positive control, the effects of KMN-159 might have been more prominent. Finally, the sample size was chosen based on previous experiments and with respect to the principles of the 3Rs allowing an initial assessment of the minimum effective dose of KMN-159 in segmental femoral bone defects. Future studies are therefore required to validate our findings.

## 5. Conclusions

Taken together, in this study, we could show a dose–dependent osteoregenerative effect of KMN-159 with reaching its maximum at 2 mg while higher concentrations did not lead to increased bone defect healing. The hypothesis stated at the beginning of this study—that KMN-159 would show a significant bone regenerative potential in a critical-sized femoral bone defect model in rats—was confirmed. The osteoregenerative potential demonstrated in our study, together with its chemical stability, good pharmacological properties, and its inability to induce unwanted ectopic bone formation [31], make KMN-159 a promising alternative to rhBMP-2. Nevertheless, further work is needed to find the optimal dose of KMN-159 required for sufficient bone healing in different bone defect models while exhibiting minimal or no adverse side effects. In addition, as has been done for other locally acting therapeutics, an optimal carrier material for KMN-159 must be established.

## Figures and Tables

**Figure 1 biomedicines-09-01712-f001:**
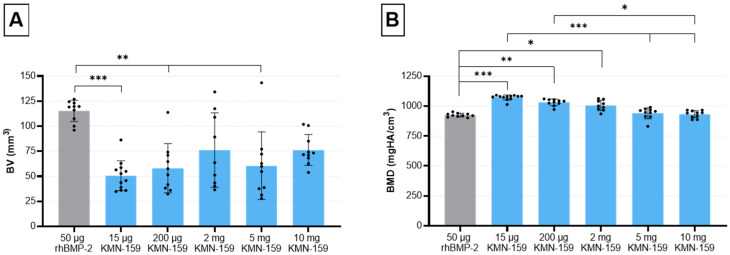
Evaluation of bone defect healing based on µCT data 12 weeks after surgery. (**A**) bone volume and (**B**) bone mineral density in the defect area (mean ± SD; Kruskal-Wallis following Dunn’s post hoc test: *** *p* ≤ 0.001, ** *p* ≤ 0.01, * *p* ≤ 0.05).

**Figure 2 biomedicines-09-01712-f002:**
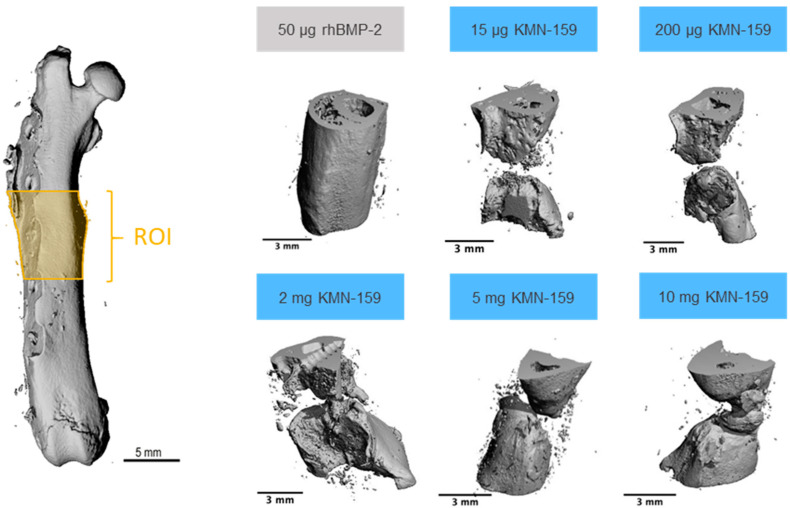
Representative examples of 3D reconstructions of the defect area (ROI) as determined by µCT.

**Figure 3 biomedicines-09-01712-f003:**
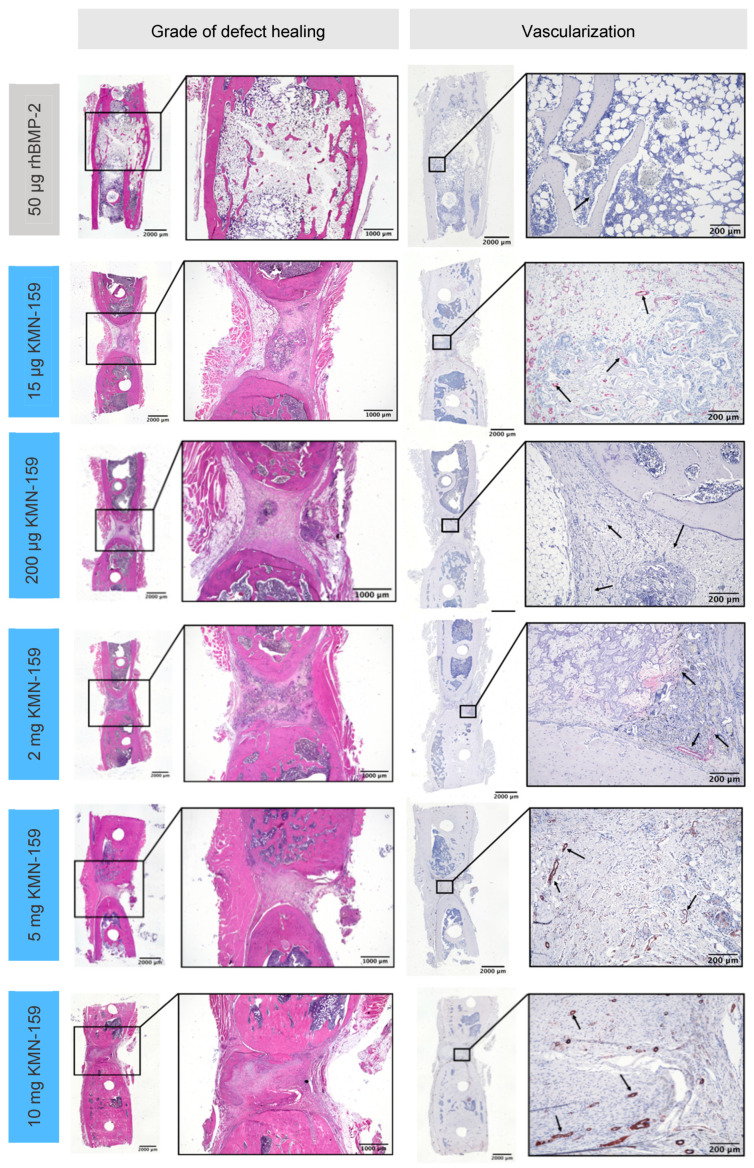
Representative histological sections for the grade of defect healing and vascularization of the defect area after 12 weeks as demonstrated by hematoxylin-eosin (HE) staining (**left**) and α-smooth muscle actin immunostaining (**right**; black arrows depict vessels).

**Figure 4 biomedicines-09-01712-f004:**
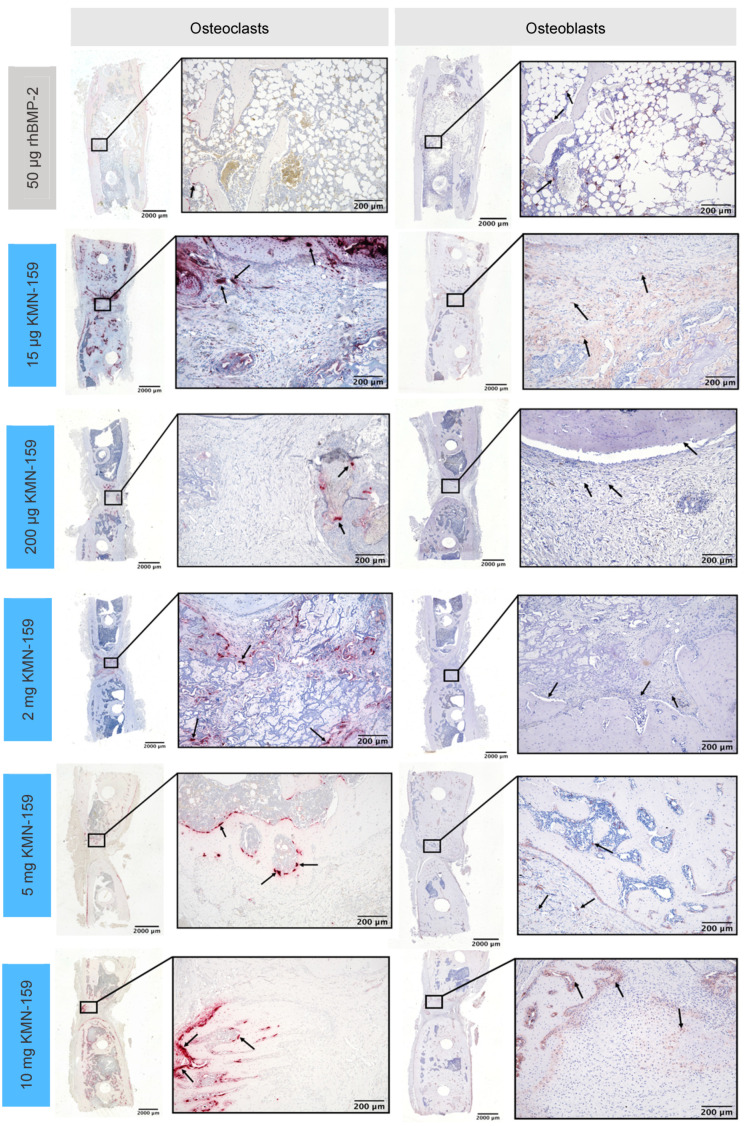
Representative histological sections of tartrate-resistant acid phosphatase-positive osteoclasts (**left**) and alkaline phosphatase-positive osteoblasts (**right**) in the defect area after 12 weeks.

**Figure 5 biomedicines-09-01712-f005:**
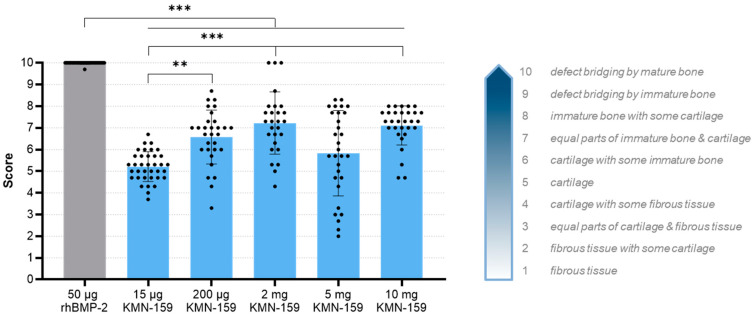
Histological evaluation of the degree of defect healing according to Huo et al. [23] 12 weeks after surgery (mean ± SD; Kruskal-Wallis following Dunn’s post hoc test: *** *p* < 0.001, ** *p* < 0.01).

**Figure 6 biomedicines-09-01712-f006:**
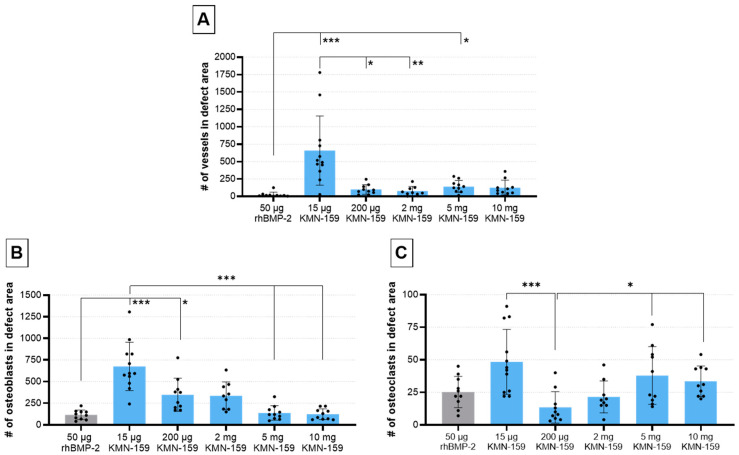
Histological evaluation of vascularization and bone cell markers 12 weeks after surgery. Semi-quantitative analysis of (**A**) α-smooth muscle-stained vessels, (**B**) ALPL-stained osteoblasts and (**C**) TSP-stained osteoclasts within the defect area (mean ± SD; Kruskal-Wallis following Dunn’s post hoc test: *** *p* ≤ 0.001, ** *p* ≤ 0.01, * *p* ≤ 0.05).

**Table 1 biomedicines-09-01712-t001:** Study overview including treatment groups, doses, and sample size (*n*).

Group	Treatment	Animals Operated (*n*)	Animals for Analysis (*n*)
1	MCM + 50 µg rhBMP-2	10	10
2	MCM + 15 µg KMN-159	12	12
3	MCM + 200 µg KMN-159	10	10
4	MCM + 2 mg KMN-159	10	9
5	MCM + 5 mg KMN-159	11	10
6	MCM + 10 mg KMN-159	11	10

**Table 2 biomedicines-09-01712-t002:** Sample size (*n*), defect bridging as well as mean value, standard deviation (SD) and 95% confidence interval (CI) of bone volume (BV) and bone mineral density (BMD) 12 weeks post-surgery as evaluated by µCT.

		50 µg rhBMP-2	15 µg KMN-159	200 µg KMN-159	2 mg KMN-159	5 mg KMN-159	10 mg KMN-159
animals (*n*)		10	12	10	9	10	10
defect bridging		10/10	0/12	0/10	3/9	1/10	1/10
BV (mm^3^)	mean ± SD	115.2 ± 10.6	50.6 ± 14.9	58.1 ± 24.6	76.1 ± 37.1	60.5 ± 33.8	76.2 ± 15.5
	95% CI	107.7–122.8	41.1–60.0	40.5–75.6	47.6–104.7	36.4–84.7	65.2–87.3
BMD (mgHA/cm^3^)	mean ± SD	925 ± 15	1072 ± 21	1031 ± 27	1006 ± 43	940 ± 46	933 ± 30
	95% CI	914–936	1059–1086	1011–1050	972–1039	908–973	911–955

**Table 3 biomedicines-09-01712-t003:** Mean value, standard deviation (SD) and 95% confidence interval (CI) of the grade of defect healing (HE-staining) and the number of vessels (α-smooth muscle actin staining), osteoblasts (ALPL staining) and osteoclasts (TSP staining) within the defect area 12 weeks post-surgery.

		50 µg rhBMP-2	15 µg KMN-159	200 µg KMN-159	2 mg KMN-159	5 mg KMN-159	10 mg KMN-159
animals (*n*)		10	12	10	9	10	10
grade of defect healing	mean ± SD	10.0 ± 0.1	5.2 ± 0.7	6.6 ± 1.2	7.2 ± 1.4	5.8 ± 2.0	7.1 ± 0.9
	95% CI	10.0–10.1	5.0–5.4	6.1–7.0	6.7–7.8	5.1–6.6	6.8–7.4
vessels	mean ± SD	24 ± 37	659 ± 497	100 ± 69	78 ± 64	144 ± 85	128 ± 105
	95% CI	0–51	343–974	50–149	28–127	83–205	52–203
osteoblasts	mean ± SD	116 ± 56	676 ± 280	349 ± 192	338 ± 159	139 ± 83	123 ± 63
	95% CI	76–156	498–854	212–486	216–460	79–198	78–168
osteoclasts	mean ± SD	25 ± 12	48 ± 25	14 ± 12	22 ± 12	38 ± 22	34 ± 12
	95% CI	17–34	33–64	5–22	12–31	22–54	25–42

## Data Availability

The primary data supporting the reported results can be provided upon request by the corresponding author.

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
