# Peer review of "Dose-Dependent Effects of a Novel Selective EP4 Prostaglandin Receptor Agonist on Treatment of Critical Size Femoral Bone Defects in a Rat Model"

_biomedicines, 2021, doi:10.3390/biomedicines9111712_

Round 1

Reviewer 1 Report

The authors are looking for an alternative to BMP in bone regeneration. They propose EP4 prostaglandin 15 receptor agonist KMN-159 and evaluate the dose-dependent effects. in my opinion, the work is very well planned and described. I suggest some minor corrections:

- chapter 2.2.: The Authors wrote „scaffolds were (…)  sterilized by supercritical CO2 [21] or by gamma irradiation”. Is it possible to clarify which scaffolds were sterilized using a specific technique? Were all the scaffolds in each group sterilized using one technique or different sterilization techniques were used in one animal group? Is it possible that the sterilization technique affects the properties of the scaffolds and, as a result, the BMP/KMN-159 release rate.

-chapter 2.2. – could you please specify how scaffolds were loaded with BMP/KMN-159. Also, how was the sterility maintained during the protein loading step (as scaffolds were sterilized before loading the proteins)

- discussion: The authors wrote „Depending on the carrier material, 36 the minimum effective dose for bridging a 6-8 mm segmental femoral bone defect in rats 37 is between 1 and 2.25 μg [33,34]. So, by using low dose rhBMP-2 as positive control the 38 effects of KMN-159 might have been more prominent.” If so, could you please explain why such high dose of rhBMP-2 was applied in the study?

Author Response

The authors are looking for an alternative to BMP in bone regeneration. They propose EP4 prostaglandin 15 receptor agonist KMN-159 and evaluate the dose-dependent effects. in my opinion, the work is very well planned and described. I suggest some minor corrections:

Comment 1:

Chapter 2.2.: The Authors wrote „scaffolds were (…)  sterilized by supercritical CO2 [21] or by gamma irradiation”. Is it possible to clarify which scaffolds were sterilized using a specific technique? Were all the scaffolds in each group sterilized using one technique or different sterilization techniques were used in one animal group? Is it possible that the sterilization technique affects the properties of the scaffolds and, as a result, the BMP/KMN-159 release rate.

Reply Comment 1:

Thank you for this legitimate question.

Due to a temporary unavailability of gamma irradiation, scaffolds of the 15 µg KMN-159 group were sterilized by supercritical CO2. The scaffolds of all other groups (50 µg BMP-2, 200 µg, 2 mg, 5 mg and 10 mg KMN-159) were sterilized by gamma irradiation. We added this information in the manuscript (page 3, lines 80 + 81).

Unfortunately, we cannot comment on whether the sterilization technique affects the release rate, because we did not investigate that issue. In general, sterilization with supercritical CO2 was shown to be less compromising on mechanical and rheological properties compared to gamma irradiation [Bernhardt et al., 2015]. Additionally, when testing cytocompatibility using human mesenchymal stem cells, viability and proliferation were not compromised by supercritical CO2 treatment. Since supercritical CO2 sterilization for the 15 µg KMN-159 samples in this study was performed under addition of H2O2 and in combination with acetic anhydride, it cannot be excluded that residues of these additives might have influenced the release rate of KMN-159.

Comment 2:

Chapter 2.2.: Could you please specify how scaffolds were loaded with BMP/KMN-159. Also, how was the sterility maintained during the protein loading step (as scaffolds were sterilized before loading the proteins)

Reply Comment 2:

In preparation of this study, the maximum loading volume of the dry sponge-like MCM scaffold was determined. MCM scaffolds with a length of 8 mm and a diameter of 4 mm as used in our study can adsorb in maximum 200 µL of liquid. Hence, loading of pre-sterilized scaffolds was done by pipetting 200 µL containing the respective amount of BMP-2 or KMN-159 onto the scaffolds. To ensure sterility, this was done under a laminar air flow bench. After loading, scaffolds were allowed to dry overnight under sterile conditions in the laminar air flow bench and then stored at 4 °C until implantation. Since BMP-2 is not very stable, BMP-2 loaded scaffolds were prepared the day before implantation. We added this information in the manuscript (page 3, lines 80-89).

Comment 3:

Discussion: The authors wrote „Depending on the carrier material, the minimum effective dose for bridging a 6-8 mm segmental femoral bone defect in rats is between 1 and 2.25 μg [33,34]. So, by using low dose rhBMP-2 as positive control the effects of KMN-159 might have been more prominent.” If so, could you please explain why such high dose of rhBMP-2 was applied in the study?

Reply Comment 3:

In this study BMP-2 was used as the generally accepted standard for inducing bone formation. We decided to use the relatively high dose of 50 µg BMP-2 to have a noticeably positive control with every defect getting bridged, as well as to get an impression of the effectiveness of KMN-159 compared to the high dose of BMP-2. Indeed, there is a point in using a lower dose of BMP-2 which would have made the study more meaningful. We will definitely consider this issue in our future studies.

References:

Bernhardt, A.; Wehrl, M.; Paul, B.; Hochmuth, T.; Schumacher, M.; Schütz, K.; Gelinsky, M. Improved Sterilization of Sensitive Biomaterials with Supercritical Carbon Dioxide at Low Temperature. PLoS One 2015, 10, doi:10.1371/journal.pone.0129205.

Reviewer 2 Report

The manuscript investigates the dose dependent effects of the bone regenerative potential of the novel selective, non-morphogenic EP4 prostaglandin 15 receptor agonist KMN-159. Different amounts of KMN-159 are loaded in mineralized collagen type-1 matrices and implanted into a 5 mm critical-sized femoral defect in rats. bone volume (BV) and bone mineral density (BMD) are analyzed to evaluate the performances, which suggest a dose-dependent effect of KMN-159 on the bone regeneration. In addition, bone morphogenetic protein 2 (BMP-2) is performed as the control experiment. The results are interesting. The paper is recommended for publication in Biomedicines. Minor revisions are suggested below. 1. The concentration of KMN-159 varies from 15 ug to 10 mg, which is quite large. The concentration of BMP-2 is only 50 ug. Should BMP-2 also show a similar dose dependence? 2. In Page 3, Table 1, why do the animals under different treatments have different number? 3. In Page 5 and Page 10, there are two Table 2.

Author Response

The manuscript investigates the dose dependent effects of the bone regenerative potential of the novel selective, non-morphogenic EP4 prostaglandin 15 receptor agonist KMN-159. Different amounts of KMN-159 are loaded in mineralized collagen type-1 matrices and implanted into a 5 mm critical-sized femoral defect in rats. Bone volume (BV) and bone mineral density (BMD) are analyzed to evaluate the performances, which suggest a dose-dependent effect of KMN-159 on the bone regeneration. In addition, bone morphogenetic protein 2 (BMP-2) is performed as the control experiment. The results are interesting. The paper is recommended for publication in Biomedicines. Minor revisions are suggested below.

Comment 1:

The concentration of KMN-159 varies from 15 ug to 10 mg, which is quite large. The concentration of BMP-2 is only 50 ug. Should BMP-2 also show a similar dose dependence?

Reply Comment 1:

In this study BMP-2 was used as the generally accepted standard for inducing bone formation. We decided to use the relatively high dose of 50 µg BMP-2 to have a noticeably positive control with every defect getting bridged, as well as to get an impression of the effectiveness of  KMN-159 compared to the high dose of BMP-2.We absolutely agree that BMP-2 shows a dose dependence which has already been  investigated and published by many others [Boerkel et al., 2011; Zara et al., 2011; Kamal et al., 2019]. In this study we focused on the investigation of KMN-159 as a possible alternative to BMP-2 and on finding a dose that is sufficient to induce and stimulate bone regeneration.

Comment 2:

In Page 3, Table 1, why do the animals under different treatments have different number?

Reply Comment 2:

We intended to have a sample size of 10 animals per group after the observation period. To compensate perioperative loss of animals due to complications, we included 4 extra animals. After 10 animals of each group had survived the operation, the remaining 4 animals were also operated and treated with the extra scaffolds available. Unfortunately, the resulting number of animals in each group was unequal.

Comment 3:

In Page 5 and Page 10, there are two Table 2.

Reply Comment 3:

Thanks a lot for this hint! We corrected Table 2 into Table 3 on page 10 in the manuscript.

References:

Boerckel, J.D.; Kolambkar, Y.M.; Dupont, K.M.; Uhrig, B.A.; Phelps, E.A.; Stevens, H.Y.; García, A.J.; Guldberg, R.E. Effects of Protein Dose and Delivery System on BMP-Mediated Bone Regeneration. Biomaterials 2011, 32, 5241–5251, doi:10.1016/j.biomaterials.2011.03.063.

Zara, J.N.; Siu, R.K.; Zhang, X.; Shen, J.; Ngo, R.; Lee, M.; Li, W.; Chiang, M.; Chung, J.; Kwak, J.; et al. High Doses of Bone Morphogenetic Protein 2 Induce Structurally Abnormal Bone and Inflammation In Vivo. Tissue Eng Part A 2011, 17, 1389–1399, doi:10.1089/ten.tea.2010.0555.

Kamal F. A.; Siahaan O. S. H.; Fiolin J. Various Dosages of BMP-2 for Management of Massive Bone Defect in Sprague Dawley Rat. Arch Bone Jt Surg. 2019 Nov; 7(6): 498–505.